# Evaluation of Clinical Parameters Associated with Response and Resistance to Cemiplimab in Locally Advanced and Metastatic Cutaneous Squamous Cell Carcinoma: A Multi-Institutional Retrospective Cohort Study

**DOI:** 10.3390/curroncol32030168

**Published:** 2025-03-15

**Authors:** Joseph Edward Haigh, Sam Rack, Ruiyang Yan, Sherin Babu, Olly Donnelly, Harriet Walter, Guy Faust, Shradha Bhagani, Patrick Isola, Robert Metcalf

**Affiliations:** 1The Christie NHS Foundation Trust, Wilmslow Road Manchester, Manchester M20 4BX, UK; 2School of Medical Sciences, University of Manchester, Manchester M13 9PL, UK; 3Portsmouth Hospitals University NHS Trust, Southwick Hill Road, Cosham PO6 3LY, UK; 4Faculty of Science and Health, University of Portsmouth, Winston Churchill Ave., Portsmouth PO1 2UP, UK; 5University Hospitals of Leicester NHS Trust, Leicester LE1 5WW, UK

**Keywords:** cutaneous squamous cell carcinoma, cemiplimab, primary site

## Abstract

Cutaneous squamous cell carcinoma (cSCC) is a common cancer with increasing incidence and 5% of patients develop incurable disease, often resistant to chemotherapy. The anti-PD-1 therapy cemiplimab has shown high efficacy in clinical trials. This retrospective study evaluated the real-world effectiveness of cemiplimab in incurable cSCC and examined factors influencing response and toxicity. Data from 86 patients across three UK healthcare providers were analysed. Median progression-free survival (PFS) and overall survival (OS) were not reached, with 38% showing durable responses beyond 12 months. The overall response rate was 60.8% (95% CI 49–71), and the clinical benefit rate was 74.3% (95% CI 63–83). A head and neck primary site was associated with improved PFS (*p* = 0.008) and OS (*p* = 0.023), while concurrent immunosuppression was associated with worse PFS (*p* < 0.001). These findings align with clinical trials, suggesting cemiplimab is effective and safe in the recurrent/metastatic setting.

## 1. Introduction

Cutaneous squamous cell carcinoma (cSCC) is the second most prevalent skin cancer, after basal cell carcinoma, and accounts for one in five skin cancers [1]. Around 48,000 new diagnoses of cSCC are made each year in England, which results in about 800 deaths annually [2]. The incidence of cSCC has risen by up to 200% in the past three decades [3], a trend projected to continue due to ageing populations worldwide and increasing awareness of this disease.

Over 95% of patients with cSCC have localised disease which is amenable to curative surgery, radiotherapy, or combination therapy. However, for less than 5% of patients, who have either locally advanced or metastatic cSCC, their disease is not amenable to curative treatments and carries a significantly worse prognosis [4]. Cytotoxic chemotherapeutics and eGFR inhibitors for advanced/metastatic cSCC were limited by significant treatment-related morbidity as well as poor durable response rates [5]. The treatment of advanced/metastatic cSCC has been transformed by the use of monoclonal antibodies targeting the programmed cell death protein 1 (PD-1) receptor. PD-1 is a cell surface receptor predominantly expressed on T cells, with an important role in the regulation of the immune system. By binding to programmed death ligand 1 (PD-L1) and programmed death ligand 2 (PD-L2) on antigen-presenting cells, PD-1 suppresses T cell inflammatory activity. Whilst useful in preventing autoimmunity, this mechanism is also utilised by cancer cells to evade immune surveillance. The PD-1 inhibitor cemiplimab blocks this interaction, enabling T cells to recognise and kill cancer cells.

In 2018, cemiplimab was approved as a first-line treatment for advanced cSCC not amenable to curative surgery or radiotherapy. This approval was based on positive results from multiple clinical trials demonstrating clinically significant and durable response rates, as well as acceptable safety profiles, in patient groups with advanced cSCC treated with cemiplimab [6,7,8]. However, a group of patients derived no meaningful benefit from treatment in these studies, highlighting a need to develop predictors of response and resistance to therapy. In this retrospective multi-institutional study, we aimed to evaluate clinical parameters in a cohort of patients with recurrent, locally advanced, or metastatic cSCC (R/M-cSCC), to identify clinical and pathological associations with the response to cemiplimab and duration of therapy and to compare with previous findings of cemiplimab efficacy and safety.

## 2. Materials and Methods

### 2.1. Patients and Treatment

All patients had a histological or cytological diagnosis of cSCC, with either locally advanced, recurrent, or metastatic disease, and were not candidates for curative surgery and/or curative radiotherapy. Patients were confirmed to be eligible for cemiplimab treatment without any contraindication or symptomatically active brain metastases or leptomeningeal disease. Patients had received no prior anti-PD-1, anti-PD-L1, anti-PD-L2, anti-CD137, or anti-cytotoxic T-lymphocyte-associated antigen-4 (CTLA-4) therapies.

A total of 86 patients were treated with cemiplimab at a dose of 350mg every 3 weeks at 3 UK cancer centres. Treatment was continued until disease progression, unacceptable toxicity as per physician assessment, or patient withdrawal, up to a maximum treatment duration of 2 years or 35 three-weekly cycles, whichever occurred first. Dose delays, toxicity management, and re-initiation of cemiplimab were at the discretion of the treating physician and consistent with the Summary of Product Characteristics. All clinical data were anonymised and no patient-level identifiable data were collected or included in this analysis; as such, no individual patient consent was required for our analysis.

Clinical characteristics including immunosuppression status, concurrent autoimmune disease, concurrent haematological malignancy, site of disease, and treatment outcomes were recorded for all patients. Immunosuppression was defined as receiving any immunosuppressive medication, including for solid organ transplantation. Overall survival (OS) and physician-assessed progression-free survival (PFS) were calculated from the date of the first dose of cemiplimab. Toxicities documented in the medical notes were extracted and defined using the common terminology criteria for adverse events version 5 (CTCAE v5).

Standard of care radiological and clinical assessments, including the choice of modality between MRI, CT, and medical photography, were performed locally as directed by the treating physician. The response was assessed based on an investigator’s assessment of radiological and clinical findings. Overall response rate (ORR) was defined as the complete response and partial response rates combined. Clinical benefit was defined as any response better than progressive disease and the clinical benefit rate (CBR) was the percentage of patients with clinical benefit.

### 2.2. Statistics

Data analyses were performed using the R programming language for Windows (v.4.4.1) in the RStudio interface (v.2024.04.2+764) [9]. Univariate Kaplan–Meier survival analysis was performed with the ggsurvfit package; *p*-values were calculated using the log-rank test [10]. Graphs were built with the ggplot2 package [11]. Toxicity odds ratios were calculated using Fisher’s exact test due to the small sample sizes. *p*-values were considered significant if ≤0.05.

## 3. Results

Data from 86 patients across three UK cancer centres were included in the analysis (Table 1). Of these patients, 72% were male and 28% female and the median age was 71 years with a range of 34 to 93 years. Prior surgery and radiotherapy were performed in 68% and 58% of patients, respectively. The primary site was the skin of the head and neck region in 65% of patients, while the remainder consisted of torso (7%), upper limb (7%), lower limb (18%), and unknown primary site (3%). Amongst the study population, local recurrence was present in 59/86 (69%) and metastatic disease in 82/86 (95%) of patients. A prior diagnosis of haematological malignancy was present in 15% of the patients, with chronic lymphocytic lymphoma being the most frequent of these (69%). Seven patients (8%) had concurrent immunosuppressive therapy at the time of treatment initiation. Three of the immunosuppressed patients were solid organ transplant recipients; all three had renal transplants (Appendix A). All three transplant recipients continued immunosuppressive therapy during treatment with cemiplimab; one of these patients started to wean off immunosuppressive therapy on confirmation of progressive disease but cemiplimab was discontinued before the wean was completed.

Of the 86 patients, 74 were evaluable for response. Of the twelve patients who were not evaluable, three patients died from non-cancer-related causes prior to assessment of response, and nine patients did not reach the first assessment scan at the time of censoring. Complete response was observed in 29.7%, and partial response in 31.1% of patients, with an ORR of 60.8% (95% CI 49–71) (Table 2). CBR was 74.3% (95% CI 63–83). The three patients with locally advanced disease without recurrence or metastases all experienced a complete response.

For the entire cohort, both median OS and PFS were not reached, and the 2-year OS rate was 54.7% (Figure 1). Durable clinical benefit lasting for >12 months was seen in 38% of patients (*n* = 33/86). At the time of analysis, eight of the complete responders had ongoing cemiplimab treatment, and the others were not undergoing treatment due to completing the maximum treatment duration of 2 years (36.3%), discontinuation secondary to immunotherapy-related toxicities (18.2%), or unrelated mortality (9.1%).

In the analysis of associations between clinical variables and PFS, patients with a primary site arising from the skin of the head and neck region had significantly improved PFS compared with other sites combined (Figure 2A). In addition, the presence of immunosuppression was associated with a significantly worse PFS (Figure 2B). For the HN primary site, the median PFS was not reached vs. 8.4 months for other sites (*p* = 0.008). For patients with concurrent immunosuppression, the median PFS was 2.9 months vs. not reached in those without immunosuppression (*p* < 0.001). For patients with a co-existing haematological malignancy (Figure 2C), the median PFS was 6.9 months vs. not reached for patients without a haematological malignancy, but this did not reach statistical significance. For patients with poorly differentiated disease (Figure 2E) the 12-month probability of progression was 45% vs. 27% for moderately differentiated disease and the 24-month probability of progression was 48% vs. 27%, but these differences did not reach statistical significance (*p* = 0.3).

In the analysis of associations between clinical variables and OS, a primary site arising from the skin of the head and neck was associated with a significantly improved OS with a median OS that was not reached vs. 9.4 months for other primary sites (*p* = 0.023, Figure 3A). Patients with a history of immunosuppression (Figure 3B) or haematological malignancy (Figure 3C) had a shorter OS, but this did not reach statistical significance. No statistically significant differences in OS were observed based on other clinical parameters including the presence of an autoimmune comorbidity (Figure 3D), histological grading (Figure 3E), or an ECOG performance status ≥ 2 (Figure 3F).

The median duration of follow up was 14 months (range 0.2–51) and the median number of cycles administered was eight (range 1–36). For the 86 patients included in the study, 47% developed immune-related (IR) toxicities of any CTCAE grade, while 13% developed grade 3 or above. Treatment was stopped in 10 patients (12%) due to immunotherapy-related toxicities. A single patient (1.2%) died due to IR toxicity.

None of the analysed clinical parameters were associated with a significantly increased risk of immunotherapy toxicity using Fisher’s exact test (Table 3). Notably, no immunosuppressed patients with a solid organ transplant were reported to experience acute graft rejection.

Three patients received palliative radiotherapy for symptomatic lesions during treatment with cemiplimab at the discretion of the treating physician. In all three cases, the decision to treat with radiotherapy was guided by symptoms and not radiological changes.

## 4. Discussion

This retrospective study demonstrated the efficacy of cemiplimab for treating patients with R/M-cSCC across three cancer centres in the UK. The 86 patients included in the study had comparable demographics to those in the clinical trial populations [6,8], although the median age in our cohort was lower than most observational studies of cemiplimab for R/M-cSCC [12,13]. A large proportion of patients were older, male, or had the head and neck as their primary site of disease. This study included a larger proportion of patients with poorly differentiated disease compared to the clinical trial population (55% vs. 28%) [8] and also patients with co-existing haematological malignancy (15%) as well as those with a performance status greater than 1 (12%), who were excluded from the Phase II trial.

An ORR of 60.8% as assessed by the treating physician was observed in the current study, including 30% with a complete response and 31% with a partial response. This is a better response compared to the 44–50% observed by Migden et al. in two clinical trials but may reflect the lack of a central assessment of response in our study [6,8]. The findings of ORR (60.8%) and CBR (74.3%) in this study are comparable to findings in other observational studies [12]. Patients with cSCC of the skin of the head and neck as the primary disease site responded significantly better than those with other primary sites combined, as reflected in the improved median OS (not reached vs. 9.4 months) and median PFS (not reached vs. 8.4 months). This result is consistent with previous real-world retrospective studies which found that the head and neck primary site of disease was associated with a longer PFS compared to the torso and limbs [13,14]. The molecular basis underlying this improved response remains unclear. One hypothesis is that disease arising within the skin of the head and neck may be associated with a higher tumour burden as a result of greater sun-exposure compared to other sites, which has previously been linked to improved survival responses to immune checkpoint inhibitor treatments in various cancers [15]. Further molecular investigations may discover useful predictive biomarkers that can help better direct treatment.

In contrast to a previous study [13], we did not observe a statistical difference in OS or PFS based on performance status; this may have been due to the smaller proportion of patients with lower performance status in our cohort.

The safety profile of cemiplimab in our cohort was comparable to other real-world studies [13,14], with 13% of patients experiencing grade 3 or above adverse events, and 12% discontinuing treatment due to adverse events. Due to the retrospective nature of this study, further details on the nature of the adverse events could not be retrieved for some patients due to insufficient toxicity data, which is a limitation of this study. In addition, no clinical parameter was associated with an increased risk of developing toxicity—most notably this included patients with an autoimmune comorbidity who are typically excluded from immunotherapy clinical trials. However, the sample sizes are small which limits the statistical confidence in the analysis.

Immunosuppressed patients were found to have a significantly worse PFS, with no patients remaining progression-free at one year after initiation of treatment. Median OS was also less than 12 months. Patients were immunosuppressed for reasons including previous solid organ transplants and the suppression of autoimmune disease. However, a limitation of this study is that full clinical details including the nature of immunosuppressive therapy for these patients could not be retrieved and the sample size was small. This paper suggests that immunotherapy is not effective for immunosuppressed patients, but further research into the association between immunosuppression and response to immunotherapy is warranted.

Our cohort included three patients with locally advanced disease without recurrence or metastases. All three experienced a complete response, but the sample is too small to draw any meaningful inferences about patterns of response in different disease stages, and further research is warranted.

Lympho-vascular invasion and the diameter of primary carcinomas are known prognostic factors of cSCC [16]. A limitation of this study is that these data were not available for all patients in our cohort and therefore have not been analysed. The potential impact of prior treatment was also not analysed as data were not available for all patients regarding the treatment intent and type of surgery, radiotherapy, or chemotherapy received prior to commencing cemiplimab.

## 5. Conclusions

Cemiplimab treatment demonstrated significant clinical benefit in patients with R/M-cSCC with an acceptable side effect profile in the real-world setting, which is comparable to previous studies. Patients with a head and neck primary site had significantly improved OS and PFS compared to those with other primary sites when treated with cemiplimab; future work should be undertaken to investigate the biological basis for this to identify useful predictive biomarkers.

## Figures and Tables

**Figure 1 curroncol-32-00168-f001:**
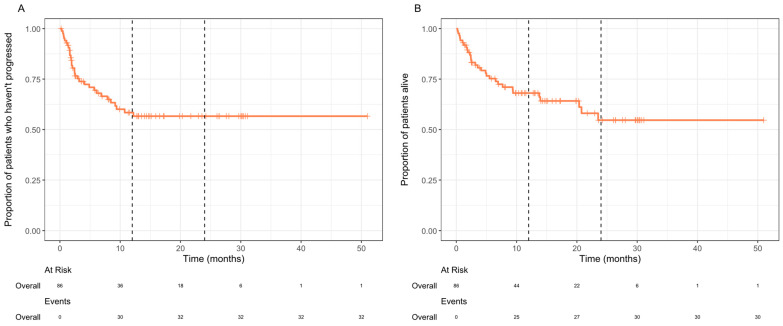
Kaplan–Meier plots of the progression-free survival (**A**) and overall survival (**B**) from treatment initiation for the entire study population (*n* = 86). The black dashed lines mark 12 months and 24 months. The median was not reached in either plot.

**Figure 2 curroncol-32-00168-f002:**
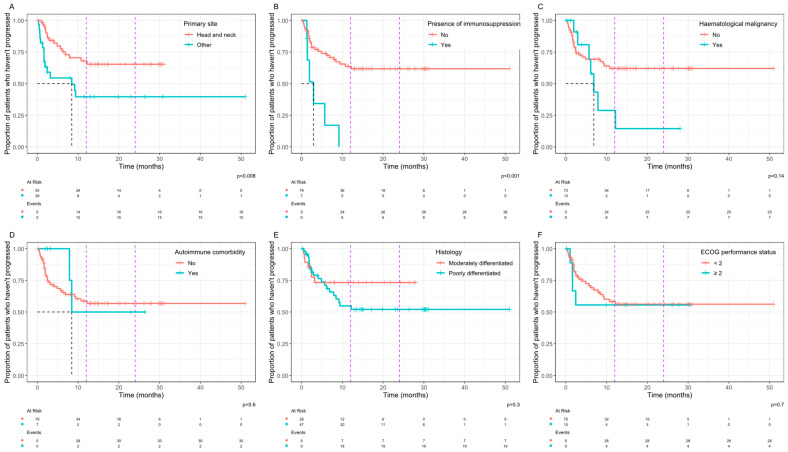
Kaplan–Meier plots of the progression-free survival with univariate comparisons of patient characteristics by log-rank, including site of primary disease (**A**), presence of immunosuppression (**B**), presence of haematological malignancy (**C**), presence of autoimmune comorbidity (**D**), differentiation on histology (**E**), and ECOG performance status (**F**). The black dashed line shows median survival if reached, and the purple dashed lines mark 12 months and 24 months.

**Figure 3 curroncol-32-00168-f003:**
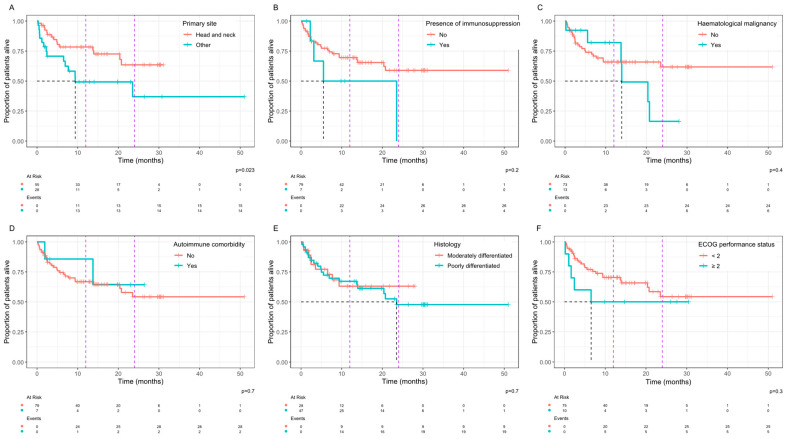
Kaplan–Meier plots of the overall survival with univariate comparisons of patient characteristics by log-rank, including site of primary disease (**A**), presence of immunosuppression (**B**)**,** presence of haematological malignancy (**C**), presence of an autoimmune comorbidity (**D**), differentiation on histology (**E**), and ECOG performance status (**F**). The black dashed line shows median survival if reached, and the purple dashed lines mark 12 months and 24 months.

**Table 1 curroncol-32-00168-t001:** Clinical characteristics of R/M-cSCC patients treated with cemiplimab.

*Characteristics*	*n*	%
**Total patients**	86	100
**Age**	71 (median)	Range 34–93
**Male**	62	72
**Smoking status**		
Current	16	19
Ex-smoker	30	35
Never smoked	30	35
Not documented	10	12
**Previous treatment**		
Surgery	58	69
Radiotherapy	49	58
SACT	5	6
**Performance status at treatment initiation**		
0	29	34
1	46	53
2	9	10
3	1	1
Not documented	1	1
**Level of de-differentiation**		
Moderate	28	33
Poor	48	55
Not documented	11	13
**Site of primary disease**		
Head and neck	55	64
Torso	6	7
Lower limb	16	19
Upper limb	6	7
Not documented	3	3
**Site of recurrence/metastasis**		
Local recurrence	59	69
Metastasis (any site)	82	95
Lymph node	64	74
Lung	28	33
Bone	16	19
Liver	5	6
Locally advanced only without recurrence	3	3
**Haematological malignancy**		
Yes	13	15
No	73	85
**Immunosuppression**		
Yes	7	8
**Solid organ transplant**		
Yes	3	3

**Table 2 curroncol-32-00168-t002:** Physician-assessed response rates. Confidence intervals calculated using logit transformation.

*Response Rate*	*n*	%	95% Confidence Interval
**Complete response**	22	29.7	20–41%
**Partial response**	23	31.1	21–43%
**Stable disease**	10	13.5	7–24%
**Progressive disease**	19	25.7	17–37%
**Overall response rate**	45	60.8	49–71%
**Clinical benefit rate**	55	74.3	63–83%

**Table 3 curroncol-32-00168-t003:** Odds ratio for developing immunotherapy toxicity based on different clinical parameters, calculated using Fisher’s exact test.

*Clinical Parameter*	Odds Ratio	95% Confidence Interval	*p*-Value
**Primary site**	0.4	0.13–1.12	0.07
**Immunosuppression**	1.58	0.25–11.53	0.70
**Haematological malignancy**	2.03	0.53–8.70	0.37
**Autoimmune comorbidity**	1.58	0.25–11.53	0.70
**Histology**	1.13	0.40–3.21	0.82
**Performance status**	1.14	0.24–5.41	1.00

## Data Availability

The data presented in this study are available on request from the corresponding author. The data are not publicly available due to the requirement to uphold the data sharing with relevant approved researchers as stipulated in the ethical approval.

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
