# Peer review of "Evaluation of Clinical Parameters Associated with Response and Resistance to Cemiplimab in Locally Advanced and Metastatic Cutaneous Squamous Cell Carcinoma: A Multi-Institutional Retrospective Cohort Study"

_curroncol, 2025, doi:10.3390/curroncol32030168_

Round 1
Reviewer 1 Report
Comments and Suggestions for Authors
Great real life experience describing the response of SCC to cemiplimab. Very relevant to current cutaneous oncology practice. My only suggestion would be that it would be interesting to publish the data of the solid organ transplant patients as there is a potential for graft rejection so I am interested to see if that happened in your cohort.
Reviewer 2 Report
Comments and Suggestions for Authors
I would recommend Major revision.
Following points:
Immunusuppression should be described.
Was it prednisolone/other immunosuppressants for pre-existing conditions, and if so for which, what dose and for how long?
Table 2 is incomplete.
The response rate is surprising, given that almost 2/3 of patients were previously treated with radiotherapy.
When was the radiotherapy? Before cemiplimab, concurrent or after initiation of cemiplimab?
"Symptomatically active brain metastases or leptomeningeal disease" is an unusual exclusion criteria in a study of cSCC.
Were organ transplanted patients included? If so, the details (which organ, what immunosuppression) must be included, in addition to whether organ rejection resulted.
The numbers in the sub-analyses are small, which statistical analysis was performed to determine how many patients needed to be included to reach valid conclusions?
Reviewer 3 Report
Comments and Suggestions for Authors
This paper is a retrospective study dealing with clinical parameters associated with response and resistance to cemiplimab in cutaneous squamous carcinoma (cSCC). It is presented well and the data are given clearly and interpreted with caution.
However, there are some points of criticism:
1. Locally advanced and metastatic cases were investigated but not separated. As there may be differences in response or resistance between these stages it would be of interest to evaluate these cases separately.
2. The histological level of differentiation is given but also the diameter of the carcinomas should be added because this is a prognostic parameter in general.
3. In addition previous treatments - surgery and/or radiation - may cause differences in response to cemiplimab and thus should be added.
4.Twenty-two patients showed complete response. Eight of them are on ongoing therapy. What were the causes for "treatment completion" for the other patients (see lane 143)?
Reviewer 4 Report
Comments and Suggestions for Authors
This is a good paper; however, several points could benefit from clarification and additional detail.
1) The sample size is small, and it is unclear whether the analysis was conducted on an intention-to-treat or per-protocol basis. Specifically, "Of the 86 patients, 74 were evaluable for response," which leaves 12 patients unaccounted for. The exclusion of these patients may have influenced the reported response rates, which are notably higher than those observed in other real-life studies (e.g., DOI: 10.3390/cancers14225543).
2) Lymph-vascular invasion is a recognized high-risk feature in the NCCN guidelines but is not mentioned in the European ones (DOI: 10.1159/000535040). Was lymph-vascular invasion assessed pathologically in this study? If so, reporting its presence or absence would add valuable context regarding patient risk stratification.
3) The role of radiotherapy needs clarification. Was it used in combination, prior to, or following the primary treatment? Providing details on how RT was incorporated into the treatment protocol would improve understanding of its impact on outcomes.
4) There is a formatting error in Table 1, where three columns appear to be misaligned or incorrectly presented.
Reviewer 5 Report
Comments and Suggestions for Authors
This is a good study with a strong statistical analysis. However, it seems that patients enrolled have a lower mean age than real life studies (Denaro N, et al. Vaccines (Basel). 2023): this could have a role in evaluating treatment response.
Furthermore, body region, immunosuppression and age are relevant clinical factors that should be taken into account as clinical predictors of response (Zelin E, et al. J Clin Med. 2022)
Round 2
Reviewer 2 Report
Comments and Suggestions for Authors
Many thanks for the clarifications.
Reviewer 3 Report
Comments and Suggestions for Authors
The revision was done according to the points of criticism.
Reviewer 4 Report
Comments and Suggestions for Authors
The authors have addressed the suggestions.